# Water-Soluble Products of Photooxidative Destruction of the Bisretinoid A2E Cause Proteins Modification in the Dark

**DOI:** 10.3390/ijms23031534

**Published:** 2022-01-28

**Authors:** Alexander Dontsov, Marina Yakovleva, Natalia Trofimova, Natalia Sakina, Alexander Gulin, Arseny Aybush, Fedor Gostev, Alexander Vasin, Tatiana Feldman, Mikhail Ostrovsky

**Affiliations:** 1Emanuel Institute of Biochemical Physics, Russian Academy of Sciences, 119334 Moscow, Russia; lina.invers@gmail.com (M.Y.); ntrofimova@mail.ru (N.T.); nsakina@mail.ru (N.S.); feldmantb@mail.ru (T.F.); ostrovsky3535@mail.ru (M.O.); 2N.N. Semenov Federal Research Center for Chemical Physics, Russian Academy of Sciences, 119991 Moscow, Russia; aleksandr.gulin@phystech.edu (A.G.); aiboosh@gmail.com (A.A.); boatsween@yandex.ru (F.G.); a2vasin@yandex.ru (A.V.); 3Department of Biology, Lomonosov Moscow State University, Leninskiye Gory 1, 119234 Moscow, Russia

**Keywords:** retinal pigment epithelium, A2E, lipofuscin granules, fluorophores, photooxidation, carbonyls, advanced glycation end products

## Abstract

Aging of the retina is accompanied by a sharp increase in the content of lipofuscin granules and bisretinoid A2E in the cells of the retinal pigment epithelium (RPE) of the human eye. It is known that A2E can have a toxic effect on RPE cells. However, the specific mechanisms of the toxic effect of A2E are poorly understood. We investigated the effect of the products of photooxidative destruction of A2E on the modification of bovine serum albumin (BSA) and hemoglobin from bovine erythrocytes. A2E was irradiated with a blue light-emitting diode (LED) source (450 nm) or full visible light (400–700 nm) of a halogen lamp, and the resulting water-soluble products of photooxidative destruction were investigated for the content of carbonyl compounds by mass spectrometry and reaction with thiobarbituric acid. It has been shown that water-soluble products formed during A2E photooxidation and containing carbonyl compounds cause modification of serum albumin and hemoglobin, measured by an increase in fluorescence intensity at 440–455 nm. The antiglycation agent aminoguanidine inhibited the process of modification of proteins. It is assumed that water-soluble carbonyl products formed as a result of A2E photodestruction led to the formation of modified proteins, activation of the inflammation process, and, as a consequence, to the progression of various senile eye pathologies.

## 1. Introduction

The retinal pigment epithelium (RPE) is a monolayer of pigment cells in close contact with the neural retina and separated by Bruch’s membrane from the choroid. RPE plays a key role in ensuring the functioning of the visual receptors [1]. RPE cells are at high risk of photooxidative stress due to prolonged exposure to light, high oxygen content, and the presence of photosensitizing pigments, including the aged pigment lipofuscin [2,3,4]. Lipofuscin in RPE cells is contained in lipofuscin granules containing lipids, minor amounts of protein, and bisretinoid fluorophores that absorb in the blue region of the spectrum [5]. Bisretinoid fluorophores are formed from all-*trans*-retinal as byproducts of the visual cycle. One of these fluorophores, A2E (N-retinylidene-N-retinyl ethanolamine) is most common in RPE lipofuscin [6,7], and its concentration in the cells can reach 20 μM with age [8]. It is important to note that fluorophore A2E accumulates in RPE cells during normal aging. A2E and other lipofuscin fluorophores are phototoxic to RPE cells, since they can generate reactive oxygen species when irradiated with blue light [9,10,11,12,13]. These data suggested that photooxidative reactions mediated by A2E and other lipofuscin fluorophores can make the main contribution to chronic oxidative stress in RPE cells [14,15]. A2E and its oxidation products accumulated by RPE cells are apparently implicated in the pathogenesis of several retinal degenerative diseases such as Best macular dystrophy, Stargardt-like macular dystrophy, Stargardt disease, and age-related macular degeneration (AMD) [16,17,18,19,20]. Therefore, it has been hypothesized that the atrophic lesions observed in retinal diseases such as Stargardt disease and AMD can be the result of poisoning due to the chronic formation of visual byproducts—in particular, A2E [21]. It was shown that A2E is capable of damaging RPE cells, the outer and inner segments of photoreceptors, and cells of the outer plexiform layer of the retina; cause the formation of sub-retinal debris; alter transcription; and diminish retinal function [21].

However, as was shown in simple model systems, the photochemical reactivity of the main lipofuscin fluorophore A2E is rather low [22]. The specific mechanisms of the damaging effect of A2E on RPE cells also remain unclear. A2E and other bisretinoids located in lipofuscin granules of RPE cells can generate short-lived reactive oxygen species directly inside the granule and therefore, the fluorophores themselves, and lipids that present in the granule become the object of their action. Indeed, products of lipid peroxidation such as 4-hydroxynonenal and malonic dialdehyde have been detected in RPE lipofuscin granules [20,23,24]. A2E accumulated in RPE cells can also form complexes with various proteins—in particular, with serum albumin [25], which can modify their photoreactivity and phototoxicity.

At the same time, some recent studies showed that lipofuscin bisretinoids are involved in the development of photoinduced glycative stress, not only in RPE tissue, but also in adjacent tissues—in particular, in Bruch’s membrane [26,27]. Chronic phototoxicity is accompanied by changes in the proteins of the cytoskeleton of RPE cells and structural proteins of the basement membrane, which is one of the characteristic indicators of oxidative modification during glycative stress [28,29]. Thus, a correlation was found between the process of photooxidation of bisretinoid fluorophores and age-related changes in Bruch’s membrane associated with the accumulation of proteins modified by advanced glycation end products (AGEs) or advanced lipoxidation age products (ALEs) [26,30,31]. In particular, in vitro experiments showed that irradiation of basement membrane proteins laminin and fibronectin in the presence of A2E leads to their modification by arginine and lysine residues [27,32]. It was found that photooxidative stress in ARPE19 cells sensitized with lipofuscin from human RPE cells caused oxidative modification of proteins and led to a significant disruption of the cytoskeleton and cell elasticity [28]. All of this indicates that reactive carbonyls formed by photooxidation of lipofuscin bisretinoids can modify molecular structures in RPE cells by forming adducts with proteins and phospholipids. Glycative stress, accompanied by the deposition and accumulation of damaged proteins, as well as the activation of inflammatory and para-inflammatory processes, significantly increases the risk of developing many age-related eye diseases—in particular, age-related macular degeneration [33,34,35,36].

It was suggested that the development of glycative stress in RPE cells is largely associated with the photooxidative destruction of lipofuscin fluorophores, during which water-soluble reactive carbonyls are formed, which are extremely cytotoxic molecules [37] and the main precursors of AGEs formation [36,38]. These carbonyls can be formed by direct oxidative decay of bisretinoids [20].

It was previously shown that the photooxidative destruction of fluorophore A2E produces a mixture of water-soluble products that can diffuse from liposomes into the surrounding incubation medium [39,40] and contain carbonyl compounds [41,42,43].

This work aimed to obtain water-soluble products of photooxidative destruction of the A2E fluorophore, and to study the reaction of protein modification by these substances under physiological conditions.

## 2. Results

### 2.1. Formation of Water-Soluble Carbonyl Products during Photooxidation of Fluorophore A2E

The RPE lipofuscin contains about 20 different bisretinoid fluorophores—byproducts of the visual cycle. A2E is one of the most well-studied by-products of the vitamin A cycle [21]. When exposed to light, A2E undergoes photooxidative destruction, which results in the formation of oxidized products containing amphiphilic and hydrophilic molecules. Photooxidation of A2E leads to the formation of various oxygen-containing products, such as, for example, ketones and aldehydes [41,42,44]. The latter ones are potentially the most toxic to the cell [37].

To determine the presence of aldehydes in water-soluble fractions, formed after irradiation of A2E with visible and blue light, we used ToF-SIMS analysis and CARS Raman microspectroscopy (Figure 1).

Figure 1A demonstrates averaged Raman spectra of the A2E samples before and after the light irradiation. It can be seen that the spectra reveal differences, including the wavenumber range of 1680–1740 1/cm associated with various C=O bands. In particular, in order to more clearly detail the appearance of specific bands of aldehydes and ketones in the oxidized A2E sample, the Figure 1A inset shows the ratio of the bands for these two samples (we divided the spectrum after irradiation with A2E by the spectrum before irradiation with A2E) in the range of 1660–1740 1/cm. In the ratio of the two spectra, a more complex spectral structure of C=O bands is visible, which include vibrational bands of 1685, ~1710–1720, and ~1730 1/cm.

The formation of oxygen-containing products (epoxides, peroxides, ketones, and aldehydes) was revealed by ToF-SIMS analysis of characteristic fragment ions containing carbonyl groups (*m*/*z* = 29—CHO^+^ ion, *m*/*z* = 43—C_2_H_3_O^+^ ion, *m*/*z* = 60—C_2_H_4_O_2_^+^ ion, *m*/*z* = 69—C_4_H_5_O^+^ ion). Figure 1B represents the accumulation of these ions during light irradiation. The intensities of all ions were normalized to the intensity of the A2E molecular ion (*m*/*z* 592), and the ion with *m*/*z* = 43 was multiplied by a factor of 0.2 for presentation purposes. As can be seen in the diagram, there was a significant increase in carbonyl ions. The ion with *m*/*z* = 60 increased particularly significantly (about 100 times). The intensities of all ions in the spectrum were normalized to the A2E intensity for every spectrum.

The results obtained by ToF-SIMS for A2E before and after exposure to light (Figure 1B) indicate the presence of aldehydes in the samples after exposure to light and correlate with literature data [41,45]. Based on the obtained structures, it can be assumed that the supernatants obtained from irradiated A2E contain a mixture of carbonyl products with amphiphilic properties.

These data were confirmed by experiments to determine the content of carbonyl compounds in the reaction with thiobarbituric acid (TBA). To do so, an analysis was carried out for the comparative content of TBA-reactive products in supernatants from A2E bisretinoid exposed to light irradiation (Figure 2A,B).

Figure 2A shows the change in the spectral characteristics of the A2E (1) after its photooxidation with blue light (2). Figure 2B shows that the supernatants of the irradiated samples contained significantly more TBA-reactive products than the supernatants of the non-irradiated samples. The obtained results indicate that, upon irradiation of fluorophore A2E, oxidized carbonyl products, readily soluble in the aqueous phase, were formed.

### 2.2. Modification of Proteins with Water-Soluble Fractions of Irradiated and Non-Irradiated Fluorophore A2E

In this series of experiments, the ability of water-soluble products obtained from irradiated and non-irradiated A2E bisretinoid to induce BSA and hemoglobin modification in the dark was investigated. It is known that reactive carbonyls, especially dialdehydes (for example, malondialdehyde or glyoxal), react with free amino groups of proteins, which leads to the formation of intra- and intermolecular covalent crosslinks and a change in the functional characteristics of the protein. This process usually takes place with intermediate formation fluorescent Schiff bases [46]. Experiments showed that incubation of proteins at 37 °C in the presence of water-soluble fractions of A2E irradiated with blue light (preparations SUPA2E-M, SUPA2E-Si, SUPA2E-LB) led to a significant increase in fluorescence intensity. This is apparently due to the modification of proteins by carbonyl products contained in the water-soluble fraction of irradiated bisretinoid A2E. Figure 3 shows that there was a linear dependence of the increase in the BSA fluorescence amplitude on the incubation time, both in the presence of water-soluble products obtained from irradiated A2E and for products from non-irradiated A2E. In these experiments, the SUPA2E-M preparation was used to modify the protein.

This is probably due to the fact that the water-soluble fraction of non-irradiated A2E also contained carbonyl products reacting with TBA (Figure 2B). However, the rate of BSA modification by water-soluble products obtained from irradiated fluorophore A2E was more than two times higher (Figure 3, curve 1).

The water-soluble products obtained by irradiation of A2E, without the use of methanol, were also active in the modification of proteins. Figure 4 shows the similarity of fluorescence spectra of hemoglobin modified with water-soluble products of photooxidative decay A2E (preparation SUPA2E-LB, curve 2), fructose (curve 3), and methylglyoxal (curve 4). However, the fluorescence intensity of hemoglobin modified by fructose and methylglyoxal was significantly higher.

Figure 5 demonstrates the increase in fluorescence of BSA when it was incubated with water-soluble products of irradiated and non-irradiated A2E (SUPA2E-Si preparation). It can be clearly seen that the irradiated products of photodestruction of A2E caused a significantly greater increase in BSA fluorescence compared to the dark control. At the same time, the increase in fluorescence in the absence of protein was insignificant (Figure 5A, curve 5). However, in the presence of the glycation inhibitor aminoguanidine, the water-soluble fraction obtained from irradiated A2E did not cause such an increase in the intensity of BSA fluorescence, and practically led to the same effect as the water-soluble fraction from non-irradiated A2E (Figure 5A, curve 4).

Aminoguanidine inhibits the Maillard reaction associated with diabetes mellitus and decreases the development of complications such as diabetic retinopathy [47]. The mechanisms of protective action of aminoguanidine are associated with both antioxidant properties [48,49] and the ability to react rapidly with dicarbonyl compounds such as methylglyoxal and glyoxal to form 3-amino-1,2,4-triazine derivatives and prevent the glycation process by these agents [50].

Fluorophore A2E exhibited a similar effect of stimulation of the albumin modification process in the dark when irradiated directly in combination with BSA. In these experiments, samples containing a mixture of BSA and A2E and no singlet oxygen quencher sodium azide were irradiated with blue light (450 nm) for an hour. Control samples were samples containing only irradiated and non-irradiated BSA, as well as samples containing an non-irradiated mixture of BSA and A2E. After 1 h of irradiation or exposure of the samples in the dark, sodium azide was added to the samples and incubated for 2 days. The results are shown in Figure 6.

It can be seen that after an hour of irradiation in a sample containing a mixture of BSA and A2E, a significant increase in the fluorescence intensity (column 1) was observed in comparison with a non-irradiated mixture of BSA and A2E (column 2). As a result of further incubation of these samples in the dark at 37 °C, this difference between light and irradiated samples was further enhanced (Figure 6A, curves 1 and 2, respectively; Figure 6B, bars 1 and 2).

At the same time, there was almost no significant difference between irradiated and non-irradiated BSA in the absence of A2E. This can be explained by the fact that when a mixture of A2E and BSA is irradiated, A2E photooxidation occurs and the products of oxidative destruction appear, which subsequently cause a dark modification of BSA. In addition, it is possible that A2E sensitizes direct BSA oxidation [25], whose products also enhance further protein modification in the dark and at elevated temperatures.

The results indicate a significant stimulation of the process of modification of the proteins (BSA and hemoglobin) by water-soluble products of photooxidative destruction of bisretinoid A2E.

## 3. Discussion

The data obtained reveal that photooxidation of bisretinoid A2E results in the accumulation of water-soluble destruction products containing reactive aldehydes and ketones. It has also been shown that these products cause the modification of serum albumin and hemoglobin, which can stimulate the development of oxidative stress in RPE cells. Oxidative and photooxidative stress is known to contribute to the development of various senile eye diseases, including age-related macular degeneration (AMD) [51,52]. It has been shown that the complex mixture of products formed during photooxidation of the lipofuscin fluorophore A2E can activate the complement cascade, leading to the development of inflammatory processes [53]. It is known that chronic inflammation contributes to the pathogenesis of many senile diseases [54], including the development of “dry” and “wet” forms of AMD [55,56,57,58,59]. However, the triggers responsible for the activation of local inflammatory responses in RPE cells remain unknown. It has been suggested [53] that one of the possible triggers of inflammation could be the photooxidation products of lipofuscin RPE.

The complement cascade is an effector system that, when activated, generates products that initiate inflammatory processes [60]. The key protein of the complement activation system (C3) is cleaved to form a small pro-inflammatory C3a fragment and a large C3b fragment. Proteins (in particular C3b) of the complement activation system are found in sub-RPE deposits and in drusen, which are a characteristic biomarker of AMD [55,56,57,61,62,63]. Moreover, it has been shown that one of the degradation products of C3b (C3d) that accumulates in tissues in which the complement system is activated is concentrated in Bruch′s membrane and in the drusen of the eyes of mice treated with modified serum albumin [64]. This indicates the stimulation of the inflammatory process by glycated proteins.

We hypothesized that water-soluble A2E photodestruction products are one of the potential sources of inflammatory signal in RPE cells. These products cause protein modification, resulting in the accumulation of defective proteins that can be recognized by the complement system as “foreign.” It can be assumed that such functionally important proteins as serum albumin and hemoglobin, when modified with water-soluble A2E photooxidative destruction products, can activate the complement system, initiate inflammation, and lead to the development of AMD upon prolonged exposure to cells. Serum albumin and hemoglobin beta 2 were indeed identified in the composition of drusen obtained from donor eyes of healthy and AMD patients, with a significantly higher concentration in AMD patients [61,65]. Thus, the interaction of proteins with water-soluble products of A2E photooxidative destruction leads to the accumulation of modified, damaged proteins. The accumulation of such modified proteins can lead to the development of inflammatory processes and importantly, underlie the pathogenesis of many eye diseases, including senile diseases in which bisretinoid A2E accumulates in RPE cells.

## 4. Materials and Methods

### 4.1. Chemicals

Sigma-Aldrich (St. Louis, MO, USA), Fluka (Buchs, Switzerland), and Component–Reagent (Moscow, Russia) chemicals; disposable plastic test tubes; and disposable Eppendorf pipettes were used in the experiments. The Sigma-Aldrich and Fluka solutions used for high-performance liquid chromatography (HPLC) were chromatographic pure.

### 4.2. Synthesis of A2E and Obtention of Water-Soluble Photooxidized Products

A2E was synthesized from all-trans-retinal and ethanolamine and purified using the methods described in [66]. The purity of A2E was monitored by HPLC on a Knauer chromatograph (Germany). The A2E concentration was determined spectrophotometrically on a Shimadzu UV-1700 spectrophotometer (Japan) at a wavelength of 430 nm and ε = 3.1 × 10^4^ M^−1^ cm^−1^.

The following procedures were used to prepare samples containing photooxidized A2E destruction products.

#### 4.2.1. Solubilization of A2E in Methanol–Phosphate Buffer Mixture

To prepare samples containing photooxidized products, we used a mixture containing 0.5 mL of A2E solution in methanol (initial concentration 1–3 mM) and 1.4 mL of potassium phosphate buffer. The mixture was divided into two identical samples. The control sample was incubated in the dark for 2 h at room temperature and with constant stirring, and the experimental sample was irradiated for 1.0–1.5 h with blue light (450 nm) from an LED source (10 mW/cm^2^) at room temperature and with constant stirring. Thereafter, the samples were centrifuged at 12,000× *g* in a Beckman Allegra 64R centrifuge for 20 min and the obtained supernatants (sample SUPA2E-M) were used in protein modification experiments and to determine TBA-reactive products. In separate experiments, for BSA modification, instead of supernatants obtained from A2E we used irradiated and non-irradiated mixtures of A2E (60 μM) and BSA (2 mg/mL).

#### 4.2.2. Solubilization of A2E In Silica Gel Suspension

The methanol solution A2E (0.7 mL) at a concentration of 2.6 mM was evaporated in a rotary evaporator until methanol was completely removed. To the remaining film was added 4 mL of 1% silica gel suspension (200–400 mesh, 60 A°, Sigma-Aldrich) in 0.1 M K-phosphate buffer and thoroughly mixed in a vortex mixer. The solubilized A2E was divided into two samples. One sample was irradiated with a blue LED lamp (450 nm) for 60 min under constant stirring and the second sample was in the dark. Then the samples were centrifuged in a Beckman Allegra 64R centrifuge for 15 min at 15,000× *g*. The obtained supernatants (sample SUPA2E-Si) were used in experiments on the modification of bovine serum albumin and hemoglobin from bovine erythrocytes.

#### 4.2.3. Solubilization of A2E in a Suspension of Polystyrene Latex Beads

This procedure was also performed as described in (4.2.2.), but 3% latex bead suspension (particle diameter 0.797 µm, Sigma-Aldrich) was used instead of silica gel suspension. The obtained supernatants (sample SUPA2E-LB) were used in experiments to modify bovine serum albumin and hemoglobin from bovine erythrocytes.

### 4.3. Measurement of Fluorescence Spectra

Fluorescence spectra were recorded on a Shimadzu RF-5301PC fluorimeter (Japan) equipped with an R955 photomultiplier (Hamamatsu, Shizuoka, Japan). For data processing RFPC software version 2.0 (Shimadzu) was used.

### 4.4. Determination of TBA-Reactive Products

The products that accumulated as a result of photooxidative destruction in the supernatant samples obtained from A2E were evaluated for the content of reactive carbonyls reacting with thiobarbituric acid (TBA-reactive products, TBARS) [67]. The concentration of TBA-reactive products was determined spectrophotometrically at a wavelength of 532 nm [68] on a Shimadzu UV-1700 spectrophotometer (Japan). The control was the original, non-oxidized samples of supernatants from A2E.

### 4.5. A2E Mass Spectrometry

Mass spectra of dried A2E solution were obtained by time-of-flight secondary ion mass spectrometer TOF.SIMS.5 (ION-TOF, Münster, Germany) equipped with a bismuth ion source. Samples were probed by 30 keV Bi3^+^ primary ions with a primary ion dose density below 4 × 10^11^ ions/cm^2^. For each sample, at least 9 areas (300 × 300 µm) were analyzed in both positive and negative ion modes. An electron flood gun was activated to avoid charging effect during analysis. The mass spectra were processed and analyzed by SurfaceLab software (ION-TOF, Münster, Germany). Ion yields were calculated as the number of ions with specific mass divided by total ion counts.

### 4.6. Broadband CARS Microspectrometry

Measurements of A2E substances in the frame of broadband femtosecond coherent anti-Stokes Raman scattering (BCARS) used a two pulse scheme for CARS generation. Narrowband pump (FWHM < 10 cm^−1^ at λ ~ 710 nm) and Stokes (FWHM ~ 500 cm^−1^ centered at λ ~ 800 nm) pulses were chosen to cover the high-frequency part of the fingerprint range (1350–1750 1/cm).

### 4.7. Measurement of Protein Modification-Induced Photooxidative Destruction Products of A2E

The process of protein modification was assessed by the formation of fluorescent Schiff bases in the reaction between free amino groups of proteins and aldehydes. Bovine serum albumin (BSA) and hemoglobin from bovine erythrocytes were used as a protein substrate. The incubation medium contained 0.1 M sterile potassium phosphate buffer, pH 7.4; 2–4 mg/mL BSA (or 2–3 mg/mL hemoglobin); 3–4 mM sodium azide; and 0.4–0.7 mL of supernatants from irradiated and non-irradiated A2E. As additional control samples, we used samples containing only proteins without supernatants, supernatants without added proteins, and samples containing a known inhibitor of the glycation process—aminoguanidine (2–4 mM). The samples were incubated at 37 °C in the dark with constant stirring for 1–3 days. After incubation, aliquots of samples were dialyzed against phosphate buffer to remove unreacted low-molecular-weight molecules. For dialysis we used a Float-A-Lyser cellulose-ether membrane (SPECTRUM Labs, United States), which allows molecules with a molecular weight of less than 3.5 kDa to pass through. Dialysis was carried out for 25 h at 6 °C. After dialysis, the fluorescence spectrum of the modified proteins was measured at an excitation wavelength of 365 nm. The content of modified proteins was estimated by the magnitude of the fluorescence amplitude of the formed products, measured at the emission maximum at 440–455 nm.

### 4.8. Statistical Analysis

The data were expressed as the mean ± SD. For the statistics, Student’s *t*-test was used. *p* < 0.05 (*p* < 0.01 for mass spectrometry) was considered statistically significant.

## Figures and Tables

**Figure 1 ijms-23-01534-f001:**
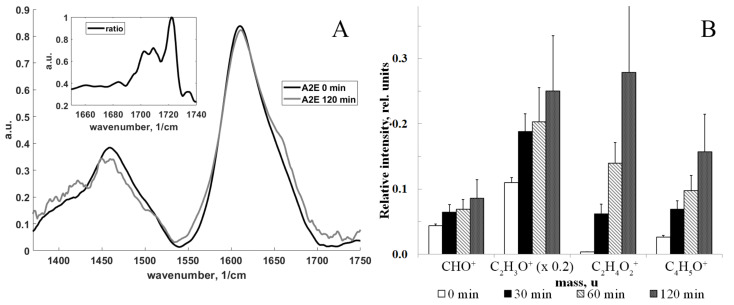
A2E oxidation under light exposure. (**A**) Raman spectra of BCARS. (**B**) Comparison of intensities of positive fragment ions containing carbonyl groups obtained by ToF-SIMS. Raman spectra data are means (means ± SD) of three assays, each with 25 spectra; *p* < 0.05. Insert in Figure 1A—the ratio of the A2E bands (after light irradiation/before light irradiation) in the range of 1660–1740 1/cm, which is the range of aldehyde and ketone vibration bands; the ion with *m*/*z* = 43 was multiplied by a factor of 0.2; ToF-SIMS data are means (means ± SD) of nine assays; *p* < 0.01.

**Figure 2 ijms-23-01534-f002:**
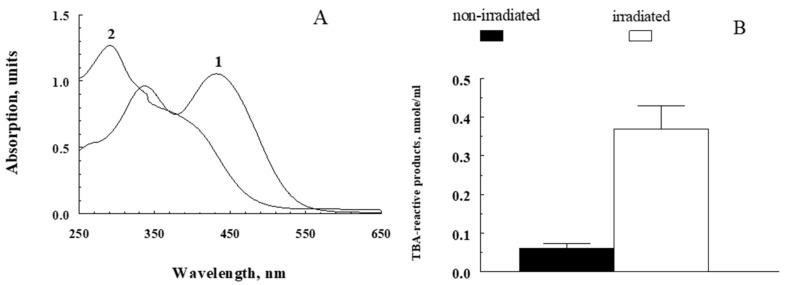
Formation of TBA-reactive products in photooxidation of fluorophore A2E. (**A**) Alteration in spectral characteristics of A2E during photobleaching (1—non-irradiated A2E; 2—irradiated A2E). The reaction medium contained 1mM A2E in a mixture of 0.1 M K-phosphate buffer and methanol (1:3; *v*:*v*). Irradiation was performed with blue light (450 nm) for 90 min. (**B**) Accumulation of TBA-reactive products in water-soluble fractions obtained from non-irradiated and irradiated fluorophore A2E. Data are means (means ± SD) of four assays; *p* < 0.05.

**Figure 3 ijms-23-01534-f003:**
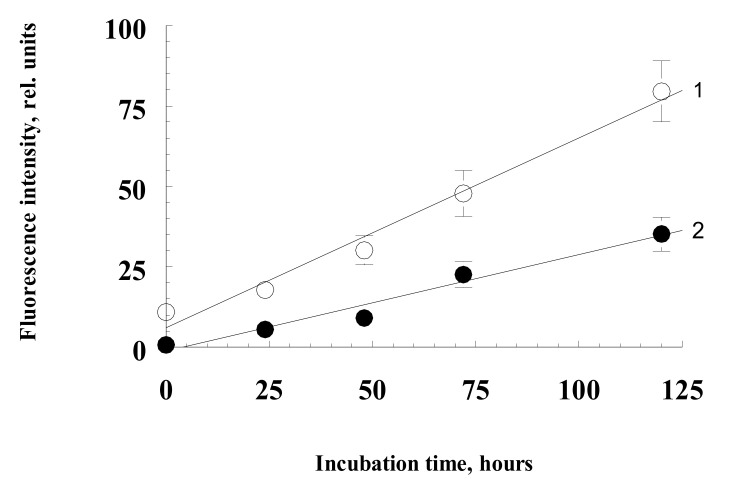
Kinetics of an increase in BSA fluorescence in the presence of water-soluble fractions obtained from irradiated (curve 1) and non-irradiated (curve 2) bisretinoid A2E.The reaction medium contained sterile 0.1 M K-phosphate buffer, pH 7.4, 2.0 mg/mL BSA, 3 mM sodium azide, and 0.7 mL of water-soluble fractions SUPA2E-M. Data are means (means ± SD) of four assays.

**Figure 4 ijms-23-01534-f004:**
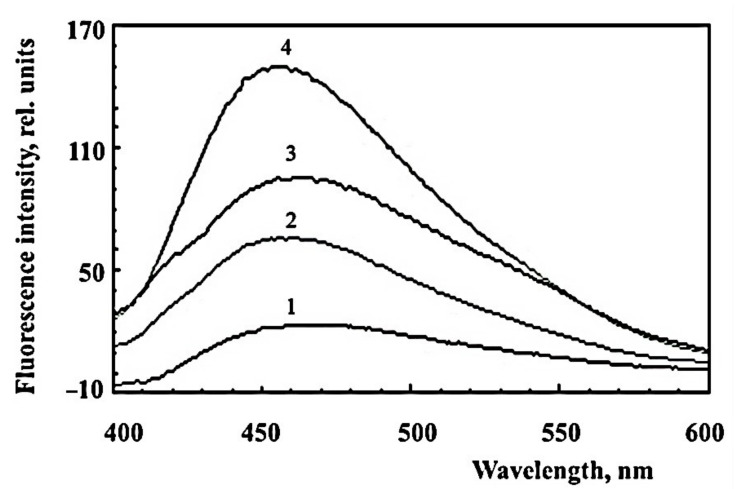
Comparative fluorescence spectra of modified hemoglobin. The incubation medium contained 0.1 M sterile K-phosphate buffer, 3 mg/mL bovine erythrocyte hemoglobin, and 3 mM sodium azide. Curve 1—300 μL of non-irradiated SUPA2E-LB was added; curve 2—300 μL of irradiated SUPA2E-LB was added; curve 3—50 mM fructose was added; curve 4—4 mM methylglyoxal was added. Samples were incubated at 37 °C for 24 h (for samples with fructose and methylglyoxal) and 48 h (for samples with SUPA2E-LB), and then dialyzed against phosphate buffer for 24 h.

**Figure 5 ijms-23-01534-f005:**
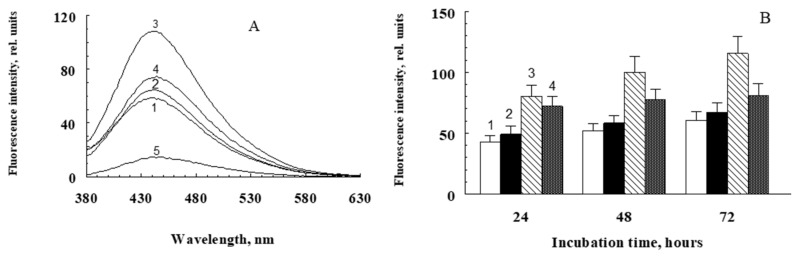
Modification of BSA by water-soluble products obtained by irradiation of A2E solubilized with silica gel. (**A**) Comparative fluorescence spectra of modified BSA after 72 h of incubation at 37 °C. (**B**) The kinetics of BSA fluorescence increased during its modification with water-soluble products of A2E photodestruction (SUPA2E-Si preparation). Data are means of three assays. The incubation medium contained 0.1 M K-phosphate sterile buffer and 3 mM sodium azide. Additives: 1—3 mg/mL BSA (only albumin); 2—3 mg/mL of BSA and 300 μL of non-irradiated SUPA2E-Si (dark A2E + albumin); 3—3 mg/mL BSA and 300 μL irradiated SUPA2E-Si (light A2E + albumin); 4—3 mg/mL BSA, 3 mM aminoguanidine and 300 μL of irradiated SUPA2E-Si (light A2E + albumin + aminoguanidine); 5—300 μL of irradiated SUPA2E-Si (light A2E, no albumin).

**Figure 6 ijms-23-01534-f006:**
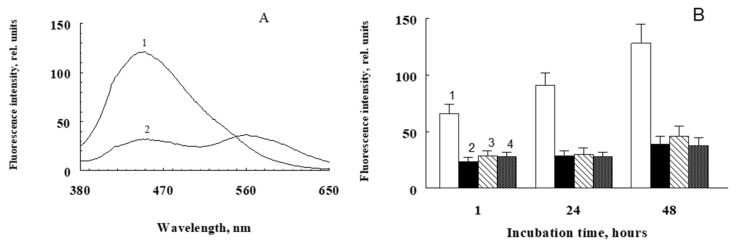
Modification of BSA upon irradiation of its complex with A2E. (**A**) Change in the fluorescence spectrum of irradiated (1) and non-irradiated (2) BSA-A2E complex after 48 h of incubation at 37 °C in the dark. (**B**) Kinetics of BSA fluorescence increase upon irradiation in the presence of A2E. Incubation medium: 0.1 M sterile K-phosphate buffer, pH 7.4, 2 mg/mL BSA, and 60 μM A2E (columns 1 and 2). The samples were irradiated with an LED source for 1 h at room temperature, after which 3 mM sodium azide was added to all samples and incubated in the dark at 37 °C. 1—BSA and A2E irradiated; 2—BSA and A2E dark; 3—irradiated BSA only; 4—dark BSA only. Data are means (means ± SD) of three assays; *p* < 0.05.

## Data Availability

The data presented in this study are available in the article and on request from the corresponding author.

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
