# Peer review of "Water-Soluble Products of Photooxidative Destruction of the Bisretinoid A2E Cause Proteins Modification in the Dark"

_ijms, 2022, doi:10.3390/ijms23031534_

Round 1
Reviewer 1 Report
The manuscript entitled “Water-soluble products of photooxidative destruction of the bisretinoid A2E cause proteins modification in the dark“ by Dontsov et al is well written and the scientific methods used are adequate for the matter under consideration, and efficiently carried out.
The results are convincingly documented, the text is clearly written without ambiguity. Moreover, the work has not been published before and the investigation could certainly have a relevance. The group around Prof. Ostrovsky is well established in the field.
The authors demonstrate with a variety of biochemical methods that A2E photooxidation formed water soluble carbonyl products which further react with proteins. Therefore this paper adds an additional piece to the puzzle around the toxicity of retinal metabolism in the retina .These modified proteins very likely are involved in cell aging and senile eye pathologies.
For these reasons, the article should be published in the international Journal of Molecular Science without any changes.
Author Response
Dear Reviewer,
Thank you very much for your careful study of our manuscript entitled “Water-soluble products of photooxidative destruction of the bisretinoid A2E cause proteins modification in the dark” (ijms- 1544440). I am very pleased with the high rating you gave to our work.
Yours sincerely,
Dr. Alexander Dontsov
Reviewer 2 Report
This manuscript describes the bi-products of A2E photooxidation and the associated protein modifications that contribute to the disease pathology in some retinal diseases such as AMD and STDG1 disease. Due to the prevalence of AMD and the destructive nature of lipofuscin, knowledge about how these granules develop and how their presence leads to toxicity is important. Most of my comments are directed toward improving the clarity of the work presented.
Title: Do you mean “bisretinoid”? I’m not familiar with “disretinoid”
Line 59, what is “LG”?
Lines 66-68. The last sentence doesn’t fit with the paragraph. Consider rewording.
Results:
Lines 102-103 the sentence “A2E as a byproduct…” does not make sense. Please re-word for clarity.
Line 110. You are looking at aldehydes but you mentioned epoxides, peroxides, ketones, and aldehydes in the previous paragraph. Why are you only looking at aldehydes? Also, in Fig. 1 you mention ketones?
Line 113. This is not a complete sentence. This entire paragraph does not read well. It is not clear what you are trying to communicate.
Line 123, what is incl. and stripes?
m/z, is this mass-to-charge? Please indicate in manuscript.
Figure 1B. The text states that the values were normalized to the intensity of A2E. Then at the end of the paragraph (line 132), the texts states that the values were normalized to the un-irradiated sample. Which one is correct? Please change to address this confusion. Also, please indicate in the legend that the m/z=43 value was multiplied by 0.2.
Figure 2B, please make the TBA-treated waveform different from the control (on the graph). Please indicate in the text which wave is associated with TBA or the control. Is un-irradiated the same as no A2E control?
Line 170, which proteins?
Why did you do the hemoglobin experiment? The text does not say anything about the relevance to include this experiment.
Fig. 5. In the legend, what is “37oC”?
Line 203, which curve is the irradiated products and which is the dark control?
The discussion is a summary of the introduction. It did not discuss the results of the experiments or how the results compare to the known literature. Moreover, there is no discussion about what the results mean with regard to disease or therapeutic strategies.
Author Response
Dear Reviewer,
Thank you for consideration of our manuscript (ijms- 1544440). I studied your remarks. It seems to me remarks quite reasonable. I would like to thank you for careful studying of our manuscript and for critical remarks. It always is useful for improvement quality of work. We made all (hopefully) the necessary changes to the text of the manuscript. We carried out a major revision of the discussion section in order to more clearly present our results, their comparison with known literature data, and their significance in relation to the mechanisms of development of age-related visual diseases.
Now regarding your comments.
Line 113. This is not a complete sentence. This entire paragraph does not read well. It is not clear what you are trying to communicate.
Thank you for the comment. The paragraph was rewritten.
“Fig.1A demonstrates averaged Raman spectra of the A2E samples before and after the light irradiation. It can be seen the spectra reveal differences including the wavenumber range of 1680-1740 1/cm associated with various C=O bands. Particularly, in order to more clearly detail the appearance of specific bands of aldehydes and ketones in the oxidized A2E sample, the Fig.1A inset show the ratio of the bands for these two samples (we divided the spectrum after irradiation with A2E / before irradiation with A2E) in the range 1660-1740 1/cm. On the ratio of the two spectra, a more complex spectral structure of C=O bands is visible, which include vibrational bands of 1685, ~ 1710-1720, ~ 1730 1/cm”
Line 123, what is incl. and stripes?
Thank you for the comment. The sentence in the paragraph has been changed to “On the ratio of the two spectra, a more complex spectral structure of C=O bands is visible, which include vibrational bands of 1685, ~ 1710-1720, ~ 1730 1/cm”
Figure 1B. The text states that the values were normalized to the intensity of A2E. Then at the end of the paragraph (line 132), the texts states that the values were normalized to the un-irradiated sample. Which one is correct? Please change to address this confusion. Also, please indicate in the legend that the m/z=43 value was multiplied by 0.2.
Thank you for the comment. Indeed, values were normalized to the A2E intensity for every spectrum.
“The intensities of all ions in the spectrum were normalized to the A2E intensity of the corresponding spectrum”
Figure 1 was updated.
All other minor remarks have been corrected in the text.
Thank you,
Dr. Alexander Dontsov

Reviewer 3 Report
The authors focused on the mechanisms of photodestruction of the RPE and Bruch’s membrane that causes various retinal degenerative disorders. They showed that water-soluble substances produced from fluorophores A2E in lipofuscin by photooxidative stress contained carbonyl compounds and they caused degeneration of BSA and hemoglobin and the anti-AGEs agent aminoguanidine inhibited the process of modification of proteins. The aim, methods and results look fine, but the composition of the manuscript is unusual. Since the result section contains statements suitable for introduction and discussion, it is confusing to know their original results and references from the literature, and discussion section described only conclusion. I recommend full revision on the descriptions of result and discussion sections.
The authors examined the modification of BSA and hemoglobin by photodestructive substances. I don’t understand how the changes of BSA and hemoglobin cause RPE and Bruch’s membrane destruction. The discussion on the relation between BAS and hemoglobin changes and retinal disorders is desirable.
Minor
Line 15 Aging → Aging of the retina
Line 24 “containing carbonyl compounds cause protein modification” Please make it clear which proteins you examined.
Author Response
Dear Reviewer,
Thank you for consideration of our manuscript (ijms- 1544440). I studied your remarks. It seems to me remarks quite reasonable. I would like to thank you for careful studying of our manuscript and for critical remarks. It always is useful for improvement quality of work. We made all (hopefully) the necessary changes to the text of the manuscript. We carried out a major revision of the discussion section in order to more clearly present our results, their comparison with known literature data, and their significance in relation to the mechanisms of development of age-related visual diseases.
All minor remarks have been corrected in the text.
Thank you,
Dr. Alexander Dontsov

Round 2
Reviewer 3 Report
Discussion usually includes the followings; the main points obtained in the present experiment, the relation between the present results and previous knowledge, significance of the present results, limitation of the present study, statement for future study, and finally conclusions. Discussion is not a review of the previous knowledges.
Author Response
Dear Reviewer,
Thank you very much for appreciating our manuscript (ijms- 1544440). and critical remarks regarding the "Discussion" section. I, of course, fully agree with your assessment of the structure of the "Discussion" section. But we, in general, have written this section in accordance with this structure. We briefly outlined the new experimental results obtained, their significance for understanding the processes of development of senile pathologies of vision, as well as a discussion of possible mechanisms for the development of eye pathologies in the light of our results. Perhaps the volume of the "Discussion" section is slightly increased, but we did not manage to write it more concisely. As for the review of previous knowledge on this issue, such a review would take more than a dozen pages of text.
Sincerely yours,
Dr. Alexander Dontsov

Round 3
Reviewer 3 Report
The authors' revision of discussion section is fair.